# Athlete Perspectives on Concussion Recognition and Management in Gaelic Games: A Qualitative Analysis

**DOI:** 10.3390/healthcare12191974

**Published:** 2024-10-03

**Authors:** Ed Daly, Lisa Ryan

**Affiliations:** 1Department of Sport, Exercise and Nutrition, School of Science and Computing, Atlantic Technological University, H91 T8NW Galway, Ireland; ed.daly@atu.ie; 2Irish Concussion Research Centre (ICRC), Atlantic Technological University, H91 T8NW Galway, Ireland

**Keywords:** recognition, concussion, Gaelic games, brain injury, injury management, mTBI

## Abstract

**Background:** The focus of this qualitative research was to interview current and retired Gaelic games athletes to ascertain how athletes viewed concussion recognition and management. **Methods:** A grounded theory methodology design was utilised to investigate concussion recognition and management experiences of a cohort of Gaelic games athletes (n = 22). Data for the study were collected using a semi-structured interview format. **Results:** Two major themes were identified: (1) an inconsistent identification of concussion as an injury and the absence of standardised procedures for removal from play, and (2) the impact of athlete-driven decision making on concussion management. Concussions were experienced by all the participants on multiple occasions while playing Gaelic games. However, concussion recognition and removal rarely occurred, as many athletes chose not to disclose their injuries or self-managed their concussions. **Conclusions:** The recognition and management of concussions in Gaelic games are currently inadequate, and they may be posing significant risks to athletes’ long-term health. This research highlights the need for more stringent protocols for concussion recognition and removal at pitch side. In addition, Gaelic games require a more objective injury management plan during acute and chronic concussion recovery.

## 1. Introduction 

Gaelic games are popular sports played in Ireland, with approximately two thousand five hundred clubs in Ireland, with a further five hundred clubs located in Europe, North America, Australia, and other territories [1]. These sports, such as hurling (males only), camogie (females only), and Gaelic football are characterised by their physicality, speed, and skill levels, making them highly popular among players and spectators alike. Gaelic games are coordinated under the auspices of the Gaelic Athletic Association (GAA), which was founded in 1884, and one of the core values of the GAA since its inception has been a pervading amateur ethos [1]. Interestingly, within the GAA, there exists ‘amateur’ athletes and ‘elite/high performance’ athletes. The category of being an ‘elite amateur’ athlete is attained by adults who represent their county (region) and play in inter-county competitions normally culminating in an All-Ireland series [1]. Regardless of whether athletes participate at an amateur or elite amateur level, injury risk exposure remains prevalent due to the physical nature of these games. 

Injuries in Gaelic games, particularly contact injuries, are common, for example musculoskeletal injures and head injuries such as concussion [2,3]. Despite growing awareness of concussion-related issues in sports worldwide, the recognition, reporting, and management of concussions in Gaelic games remains highly insufficient [4,5]. Concussions are traumatic brain injuries that can occur when an athlete receives an impact to the head or to their body [6]. In addition, it is now widely accepted that female athletes may be more susceptible to head injuries when compared to male athletes, and they may experience concussions via different mechanisms of injury, such as a whiplash-type injury with an impact to the ground or playing surface [7]. Common symptoms can range from dizziness, blurred vision, cognitive impairments, and, at times, emotional or gastrointestinal symptoms [8,9]. In Gaelic sports, there is some evidence to suggest that concussion knowledge exists within the GAA; however, many incidences of concussion are largely underreported, and concussions often go undiagnosed or misdiagnosed [10,11]. 

There are many challenges in relation to tackling concussions within Gaelic games; one of the primary issues is the lack of a standardised set of educational tools for concussion awareness amongst its coaches, athletes, and referees [11]. Another anomaly in the GAA is the difference between adult male and adult female return to play protocols in terms of the stand down from activity once a concussion has been formally diagnosed. For adult males, the stand down time is seven days, and for adult female athletes, the stand-down period is fifteen days [12]. 

As most Gaelic games are primarily amateur by nature, consequently, the availability and presence of suitably qualified medical staff may be absent or minimal [13]. These circumstances can invariably lead to many injuries, including concussion, being overlooked or not being adequately assessed at the time of injury or in the acute phase of injury [14]. As is common in many contact or collision sports, there is a culture of resilience which may not create an environment that encourages athletes to report their symptoms, particularly when reporting an injury that may affect playing time prior to perceived important games [13]. The GAA have made some efforts to address the issue of concussion in their sports via guidelines for youth and adult athletes and some educational resources to the broader GAA community [1]. However, the implementation of these guidelines and resources is highly inconsistent with limited to low adherence rates, and there are no accurate data regarding the true incidence of concussion within their organisation [13,14].

The GAA promote the use of the Sports Concussion Assessment Tool (SCAT 6) tool for medically trained personnel, and the Concussion Recognition Tool 6 (CRT6) for concussion identification. In addition, they endorse the use of a vestibular/ocular motor screening (VOMS) tool for baseline testing, identification, and management. These initiatives are available for amateur and elite amateur teams; however, the RTP protocols are not strictly followed and can increase injury risk exposure to athletes who remain symptomatic [13,14]. 

The failure to properly recognise and standardise the management of concussions may have implications for further injury risk to the long-term health of athletes [15]. There are multiple short-term consequences of under-managed concussions, which may include cognitive disfunction, an absence from education or employment, prolonging the period of recovery, and an increased risk to secondary or recurrent injury [16]. In addition to long-term health implications, there may be potential ethical or legal considerations. As has been demonstrated in sports such as rugby union, sports organisations are not immune to liability issues where inadequate concussion management protocols were not established and/or not adhered to for the welfare of their athletes [17]. From another viewpoint, sports organisations have a duty of care to their athletes to protect them from harm; this duty of care may be compromised if there are not robust concussion protocols or if these protocols are not enforced [18]. 

An additional level of complexity, which occurs in many sports, is the marked differences observed in concussion management between male and female athletes. The added complexity arises for many female athletes, as they may have less availability to medical attention and resources in Gaelic sports [19]. This unavailability of suitable concussion management tools may lead to delayed diagnoses and treatment [20,21]. 

This research sought to understand the current state of concussion recognition and management in Gaelic games, and how these guidelines are impacting current concussion management practices from the athletes’ perspectives.

## 2. Methods

### 2.1. Study Design

This study used a grounded theory methodology [22] design to investigate current and retired GAA athletes’ perspectives on how they perceived concussion as an injury in their respective sporting code. This research explored their perceptions of how concussion was recognised and managed by team mentors and medical staff within Gaelic games (i.e., hurling, camogie, Gaelic football, and ladies’ Gaelic football). Data for the study were collected using a semi-structured interview format [23]. All participants have played Gaelic games as an athlete, either at an amateur or elite amateur level, and some have retired from their code (n = 4); however, the majority of participants interviewed remain actively involved in playing Gaelic games at an amateur or elite amateur level (n = 18). 

The interview questions were designed to stimulate responses on their career to date that would clarify their opinions on concussion recognition and management within Gaelic games. The research team actively sought a broad spectrum of male and female athletes from various regions (counties) in Ireland, in order to offer a set of comprehensive perspectives. 

### 2.2. Ethics and Procedure

Ethical approval was granted for this research via the Research Sub-Committee of Atlantic Technological University. The initial group of participants was identified via existing contacts from the lead researcher seeking participants for the study. The initial pre interview meetings were an opportunity for prospective participants to discuss the study and how the recorded information would be managed in a confidential manner. Data were collected during online interviews (via Microsoft Teams) ranging from twenty minutes to sixty-one minutes in duration, where a precursory discussion was carried out with each participant to clarify the rationale for the study and how the study would commence and proceed using the semi-structured interview format. Prior to the interviews, each participant was sent a participant information sheet seeking written consent prior to their interview. Their verbal permission was sought and granted in addition to the written permission, once the participants had no further clarifying questions, the interviews proceeded using a standardised series of questions, which were used consistently across all interviews. At the conclusion of the interview, it was further clarified that all information discussed would be treated as confidential and fully anonymised for the purposes of this research. 

### 2.3. Participants’ Characteristics 

The participants in this research had all played Gaelic games at various levels throughout their careers, with most participants (n = 18) remaining active at either amateur or elite amateur levels during the timespan of data collection. The following regions were represented: Tyrone (n = 3), Clare (n = 7), Tipperary (n = 1), Cork (n = 1), Limerick (n = 1), Galway (n = 4), Wexford (n = 2), and Mayo (n =3). From the full group of participants (n = 22), 18 had represented or currently represent their respective counties (elite amateur) at an adult senior level in their respective code (82%). The participants were composed of fourteen males (64%) and eight females (36%), with an average age of 31 (SD ± 5) years old and 28 (SD ± 5) years old, respectively. All participants in the group had experienced at least one medically diagnosed concussion. Many participants were unsure with regards to the total number of concussions they have experienced, either diagnosed or undiagnosed. Within the participant group, 14% had retired due to concussion, and 55% reported long-term concussion symptoms, or experience persistent symptoms after the initial concussive event.

### 2.4. Sampling and Eligibility Criteria

This research study used an exponential non-discriminative snowball sampling method [24], where the first participants recruited were able to provide referrals. Each new participant was interviewed until data saturation occurred. Participants were continuously notified that they were under no obligation to provide any additional participants for the study.

All participants were either current or retired Gaelic games athletes who provided personal referrals to athletes who were contacted regarding taking part in this research. The semi-structured interviews used in this research enabled a standardised set of responses from the participants. This enabled the researchers to identify common emerging themes from the cohort of athletes. In doing so, the players illustrated their experiences of being a Gaelic games athlete, and their accounts of concussion recognition in their chosen code (i.e., hurling, camogie, ladies’ Gaelic football, and male Gaelic football). 

### 2.5. Data Analysis

#### 2.5.1. Transcription

The MS Teams (Microsoft, Redmond, WA, USA) recordings generated a transcript of the interviews; these transcripts were initially checked and transcribed by the lead researcher. These transcripts were scrutinised for accuracy against the original MS Teams video recordings and a secondary audio recording to clarify audio errors and further edit the accuracy of the transcripts (n = 22). 

#### 2.5.2. Decontextualisation

The process of the decontextualisation of each interview enabled the lead researcher (ED) to become more familiar with the overall content of each individual transcript. The process involved listening to the audio recordings in tandem with reading each transcript to confirm that the questions and responses to the questions correlated accurately with the audio files of the interviews; this was an iterative process that was performed on 4–5 separate occasions depending on the level of editing per interviewee. 

#### 2.5.3. Recontextualisation and Categorisation

The lead researcher identified themes, categories, and subcategories for the twenty-two athlete interview transcripts according to the method of Braun and Clarke (2019) [25]. During the recontextualisation process, the original list of identified themes was examined, and they were amalgamated as subcategories to reduce the overall number of themes [26]. Table 1 displays the athlete perceptions of concussion recognition and management from this group of Gaelic games athletes. 

### 2.6. Coding

Coding for this research utilised a semantic interpretation of the interview transcripts [27]. This process was conducted by the first author (E.D.) and reviewed with the other member of the research team (L.R.). The initial codes encompassed the entire spectrum of experiences of concussion recognition and management practices from the athlete’s perspective in Gaelic games.

Themes were identified by the first author (E.D.) in discussion with another author (L.R.). Potential themes such as inconsistent concussion recognition and the impact of athlete-driven decision making on concussion management were identified. Following this process, the authors established categories to organise the data into common groups. Subsequently, subcategories were established to represent the shared relationships between the categories. Two overarching themes were identified following an additional analysis of the sub-themes (see Table 1).

## 3. Results

In this study, two major themes were identified: (1) the inconsistent identification of concussion as an injury and the absence of standardised procedures for removal from play and (2) the impact of athlete-driven decision making on concussion management. These themes were further classified into categories and subcategories (see Table 1).

### 3.1. Theme 1—The Inconsistent Identification of Concussion as an Injury and the Absence of Standardised Procedures for Removal from Play

#### 3.1.1. Absence of Standardised Processes and Injury Management

During the interviews, it was the apparent that the group of participants believed that there was an inconsistent approach to the recognition of concussion as a recognised injury in Gaelic games. This was strongly supported by the participants, as they believed concussions were dismissed as a non-injury within Gaelic games. This was evident whether the athletes were involved in amateur Gaelic games or elite amateur Gaelic games. *“I would say they (concussions) aren’t taken seriously as a broken hand as a pulled hamstring or something like that because you can’t like run because the hamstring is pulled, so you know he’s gonna be two to four weeks”* (P2) and *“I was severely concussed but there was like, I don’t think anyone from a medical perspective who came and suggested taking a few days off before return to play protocol”* (P4), respectively.

This approach was consistent across male and female Gaelic games; one of the female participants observed: 


*“That’s probably the biggest thing I think, the next steps after you’ve been concussed is where they (management team/medics) all fall down. And they’ll know to take you off, they all know it’s concussion. They can all say go to the doctor, but like they don’t really know how to help you going forward”*
(P10).

Some participants stated that fellow athletes were withdrawn from the field of play, as their personal performance during the game had declined. The decline in performance was not seen by the management team as a concussion, it was deemed that the athlete was not playing well. However, their fellow teammates viewed the circumstances differently: 


*“I unfortunately sat beside a player after coming off, Clare versus Limerick, and I was sitting beside the player that came off and he couldn’t remember his name, he couldn’t remember how he scored. We informed him of the answers and then ten seconds later he asked the same ones again”*
(P11).

A common component of the interviews regarding the lack of recognition was that players remained on the field of play while symptomatic; this, in many instances, led to further injury exposure:


*“I played on into half time and then at half time we went into the dressing room, and I remember complaining of a headache….I went back in the second half then and as the second half of get going on, I was getting worse than for finish I couldn’t even see. I couldn’t see the ball or couldn’t hardly see it all in front of me. And then at the end I kind, I just collapsed at the edge of the pitch”*
(P12).

It may be argued that amateur settings are more prone to concussion not being detected or recognised in a timely manner due to the basic provision of medical assistance. In extreme cases, this can have far-reaching effects for amateur athletes when there is an absence of pitch-side medical assistance *“I like lapsed into a coma on the edge of the pitch…. I went into Limerick Hospital; I started having seizure. So, the doctors put me in an induced coma, and I was on life support for ten and half hours”* (P12). One participant summed up the general approach to the general lack of recognition in the following statement: “*I actually had a great chat with one of the neurologists and he said that he sometimes wished that they (the players)get another injury on top of the concussion because they’ll give their brain (some) time to recover”* (P1). 

#### 3.1.2. Lack of Action or Positive Action from Medics and Physios

The consensus from the participants was that many concussion incidences, either their personal incidences or those of their teammates, went unreported. In many cases, the level of care was dependent on the organisational structure in which the player was located. This was consistent across amateur and elite amateur organisations, where some participants believed the processes were adequate, and some fell short of what was deemed as basic requirements: 


*“I would say it kind of depends on the environment you’re in, like I said, in my club we took it seriously because one of our players got an injury (concussion), he had to retire. Like what I would say, about sport in general its amateur sport anyway, where you don’t have your full-time physio or full time and S&C”*
(P2),


*“I was bleeding from my ear or bleeding from somewhere else deeper and I just went off the field because I was groggy and remember lying down in the dugout. We didn’t have a doctor.......I remember lying on the bench and I remember not being able to see......maybe struggling to focus on the colour of jersey”*
(P3),

and, conversely, *“Our medical team, are extremely sharp, I’ve actually seen players being taken out against their own will and their own accord and so that’s been really good”* (P4). 

It was apparent from the participants where they had experienced multiple concussions, as the team medical staff were more vigilant and conscious of an athlete’s concussion history: *“This time around (after a series of concussions), the doctors, the physios were very cautious….or expertise or testing that needs to be done, were done”* (P6). Some of the participants believed, particularly in cases involving female athletes, that the initial assessment and treatment was not suitable in their playing environment: *“Actually, it’s frightening enough that it’s quite similar across every level (club and county) and it’s (pitch side concussion assessment) maybe not fit for purpose”* (P13). 

This point is connected to the level of follow-up treatment and communication from the medical staff with the injured athletes. Acceptable levels of interactions post injury varied greatly; some athletes described ongoing and consistent treatment plans: *“They (medical staff) would have been talking the coaches and then the doctor would have made contact with you later that evening or else the next morning to see if there were any might side effects”* (P3), whereas other participants experienced a different level of care: *“But I never got that checked to see if it was concussion. It was a club match, so we don’t have and for small club, but we don’t have the kind of physios and doctors and stuff”* (P16). 

Certain athletes held the belief that individuals managing concussions in Gaelic games do not have sufficient training on how to deal with a head injury: *“I probably don’t think it’s treated equally anyway, like as in I think the biggest problem is probably the practitioners working with the teams aren’t in a position to deal with it (concussion)”* (P10) or *“We didn’t have that kind of expertise to deal with (concussion) like our doctor was a nice guy but he wasn’t good enough”* (P6).

A commonly held point of view from some participants was related to the aftercare available in emergency room settings (hospitals). Many believed that these settings do not have sufficient knowledge for recognising the symptoms of concussion or offering a basic level of advice to the injured athlete, for example 


*“I got discharged from the Limerick Hospital the next morning. I was given an IRFU pamphlet on how to deal with concussion and that was it, so I think that in some way sums up how the GAA deals with it, they kind of pass the buck to maybe the team medics or the IRFU”*
(P5),

which was supported by *“The last time I got to the hospital in an ambulance (I was) there for hours and hours and then eventually they gave me a sheet of paper, and said if any of these symptoms come back, come back in”* (P6). 

#### 3.1.3. Non-Standardised Baseline Testing in Elite Amateur Athletes

The current circumstances with the GAA allow for all elite amateur athletes to have access to a baseline concussion testing. Whether these services are availed of by each county organisation is up to the discretion of each individual county. It was apparent during the interviews that most participants had not received a baseline test for concussion injury management purposes. The utilisation of these baseline services was determined by the teams’ medical staff, and they were not compulsorily imposed by the GAA. Where baseline testing was used in elite amateur settings, the impression was positive, as it removed the athlete from the decision-making process *“We have a baseline record of concussion and then we do a bit an assessment, a concussion assessments like computer thing and then he’ll tell us if we’re ruled out or not*” (P17). There were discrepancies observed in female elite amateur settings with respect to the administration of baseline testing: *“Unless you’re in a men’s setting and I think it tends to be looked after a little bit better there, probably because they’ve team doctors that are a little bit more aware of it”* (P10) or “*People don’t really talk about us (females)…. so, I think that’s probably more of it, not that it (concussion) doesn’t happen, but people still don’t really acknowledge us (females)”* (P13). In some instances, the presence of an objective concussion test confirmed what the medical staff suspected, and in doing so, facilitated the athlete to access further specialist treatment to assist the recovery process: *“It took me longer to come back this time as I was working with a physio in the Bons there in Galway and she kept failing me, so she wouldn’t let me back to contact sport until I passed that the test”* (P15) and *“I went to see the specialist and after trying to get (an appointment), he did the cognitive test again, and I actually did better than the first time I did it. So that there was nothing wrong with me cognitively”* (P16). These examples further underline the necessity for an objective head injury assessment and a holistic approach to medical care and reassurance for the injured athlete. 

### 3.2. Theme 2—The Impact of Athlete-Driven Decision Making on Concussion Management

In tandem with the reported disparity of concussion recognition in amateur and elite amateur settings from an athletes’ perspective, a second identified theme was the influence of athletes themselves on the decision-making process regarding concussion identification and management. 

#### 3.2.1. Influencing Medics, Non-Disclosure, and Being Dismissive of Symptoms

Due to the highly competitive nature of sport regardless of whether it is at an amateur or elite amateur level, athletes will attempt to manipulate removal from play decisions. This position was highly obvious based on comments from the participants: *“I would have argued black was white with the doctors to get to play on and like they (were) saying you need to come off“* (P3) and “*I was more or less nearly knocked out…..when you went down the physio would run out on the field, just be like a time-wasting exercise, you know? Take off the helmet, take your breath”* (P2). It was a common practice for all participants, irrespective of the level or importance of the game being played, or to the extent of not missing a training session in advance of an important competitive fixture: *“I was back training two days later....it wasn’t even discussed about taking it easy or, you know, we need you to take a rest period and we had the Ulster final seven days later, (I) played in it”* (P4). 

Many players dismissed any medical attention to remain on the field of play, even though they had just experienced a concussion: *“I was 100% out for the count I could hear just echoes in my head for five minutes and yeah, like I remember I was marking _______ and I was like, I was just looking for a yellow helmet. I was seeing blurred vision”* (P6) or 


*“I did talk to my team doctor, and he was just like at the time, he really didn’t think it was a concussion or not, but looking back I think it was….but yeah, I played on. I played on the rest of the game”*
(P21).

During the interviews, it emerged that many of the athletes were somewhat dismissive of concussions or concussive symptoms before they had experienced the injury or multiple concussions. Some participants were of the opinion that this viewpoint was engrained from an almost cultural origin, where they perceived this phenomenon as the following:


*“a society thing or something, it’s very much, I don’t want to be seen to be the player to come off the field like it’s not something that you can see a broken leg or blood gushing from somewhere. It’s very much a hidden injury, even when I suffered from it”*
(P11)

and “*Club managers they’re a bit backward but like if you said to a club manager, ‘oh, geez, concussion, I’m not training tonight’…… the club manager would be like ‘your grand like, you know, it’s only a belt in the head’”* (P20), or “*It might have been nearly seen as if you’re being a bit soft, there was nearly a narrative there that it isn’t that serious even though it actually is”* (P22). This may indicate a broader cultural shift of attitudes towards concussion as an injury, which would require concussion to have the same recognition as other common injuries associated with Gaelic games. 

#### 3.2.2. Diagnosed and Undiagnosed Incidence of Concussion

Being cognisant of the existing culture in Gaelic games with respect to concussion awareness and education according to the participants, it stands to reason that how athletes describe their experiences and symptomology reflects a general malaise towards concussion as a notable injury in Gaelic games. As previously mentioned, many of the athletes involved in this study attempted to conceal their concussion or concussion symptoms to compete. Interestingly, they used various different words to soften the extent of the injury, such as ‘knock’, ‘bang’, ‘little episodes’, ‘major bang’, ‘delayed concussion’, ‘proper ones’ or ‘groggy’. The overriding view was that many never truly considered concussion as an injury until they had personal experience of the debilitating effects of a brain injury: *“You don’t know, you’d never actually really comprehend it until you’ve either suffered it yourself or actually witnessed it first-hand”* (P11) and “*I think it’s like most things, whenever it visits your own door, you become a bit more curious about it and aware of it”* (P4), or “*With the crying, that evening. I didn’t even link that it was a concussion until a couple days later. I had a headache that night and I was sick that night”* (P14). 

This point links in with the low knowledge levels of athletes and the lack of recognition from team management:


*“My first one like, it was nearly like there’s nothing wrong with you, and I didn’t believe there was, the management didn’t believe. I don’t even know if we had a physio, I think it was just one the selectors and a bottle of water”*
(P9)

or *“I think they (team management) were just more so fascinated by it (concussion) because, again, like me, they didn’t really know anything about it”* (P16). 

A noteworthy opinion was if the athlete was not asked, or was able to conceal the injury, this was generally accepted at face value by the management and medical staff: *“You can suspect by someone the way they’re going on that, ‘Geez, they’re not right’. But if they’re telling you, and you look half OK, you kind of might just let it slip (not disclose symptoms)”* (P22). Many players failed to recognise the full effects of concussion, even though they may have had some knowledge of the symptoms: 


*“I’d say I had the same experience that a lot of players, I had concussions and at the time I thought that was all concussion had to offer. You know, like a couple of days of feeling out of sorts, some headaches, things like that, some memory issues or cognitive issues on the day or the day after”*
(P5).

Even though athletes, knowingly or unknowingly, chose not to disclose concussions, the diligence of knowledgeable medical staff regarding concussion managed to identify concussive incidents: *“They don’t tend to mess about like, the doctors are pretty strict and regimented on the aspects and the dangers of it”* (P3) and “*Our medical team, are extremely sharp, I’ve actually seen players being taken out against their own will and their own accord and so that’s been really good*” (P4). 

Based on the comments of the athletes, it was relatively common for athletes to play even though the management knew they were symptomatic: *“He (coach) told me about a time that he let a player go back on the pitch, although he knew she stumbled”* (P17). On some occasions, these actions were linked to the perceived importance of the game being played:


*“To be fair, the manager, did have a chat with me and didn’t start me in the Ulster final because he knew I wasn’t right (symptomatic), and he put me on as we were getting beaten, the first ball I to go for, I took a knee to the temple area”*
(P1).

On other occasions, the athletes were displaying physical signs of concussion, which reinforced the symptoms: *“I think I went to play on, and I turned around and vomited all over the pitch, so that got me taken off and I had a black eye, which I think was why people were more conscious of it”* (P13) and *“I was removed because my ear was bleeding and blood stopped and then by the time I was ready to come back on again. I couldn’t see, so they didn’t put me back on”* (P3). 

Due to multiple experiences, or long-term symptomology, many of the participants began to accept the gravity of being concussed. These cumulative effects and tolerance to concussive impacts manifested in various ways: “*First time it made it took five minutes for you to lose your vision. Second time made it like it would have taken less and less (time to feel symptomatic)”* (P3), “*You’re more susceptible to them (concussions), and more sensitive to any bangs in the head or a collisions as well or any concussion”* (P4), and “*I didn’t get another major bang that year and then the following year, I got another bang and that kind of, yeah, that kind of was the beginning of the end”* (P12). 

#### 3.2.3. Athletes Determining Length of RTP and Self-Managed Recovery

Some participants stated that they progressively moved through what they considered a well-monitored RTP. In most cases, the athlete self-determined when they returned to activity based on how they were feeling without a full understanding of their physical condition. It stands to reason that these circumstances were demonstrated due to a non-disclosure of symptoms, a shortfall in knowledge regarding head injury, the lack of a clear education program about concussion, or by a self-classification of the injury themselves: “*In general, it was no real awareness around it. I didn’t know that there was such a thing as long-term effects of it…. you shake it off and you play on*” (P9), *“I kind of just glossed over it like it was a thing that wasn’t monitored, it wasn’t a thing that they were checking*” (P10), or “*But even after the first one, it didn’t feel like there was any rules around us with the GAA regarding RTP)*” (P18).

What did occur in most cases was that the athlete came back to activity when they ‘felt’ they were able to take part in training sessions or they wanted to make themselves available on game day. In many instances, the RTP process was self-managed without consistent consultations with the team management or the team medical staff: *“You would have just sort of stepped back in (after a concussion without medical clearance) and the strange thing was, that management thought because you were that fit, that you’re ready to go”* (P1) and


*“One of the senior girls had actually said ‘you’ve concussion’, and I know she had passed that (information) on to management, but no one from management actually checked up on me until three weeks later, to find out why I wasn’t at training”*
(P14).

These reported behaviours raise an unseen, yet essential question about how RTP was being managed. Where there appeared to be a substantial disconnection between the recovering athlete and the management team, who was responsible for the athlete’s recovery. Similar to the athlete determining the duration of the RTP, it was apparent, in the bulk of cases, that the burden of managing the chronic phase of the injury fell to the athletes: *“I started going to physio (specialised in concussion treatment) that I started to know minimal improvements for the first few weeks, but then drastic improvements there on afterwards and so it wasn’t until I actually started going to physio that I noticed the improvements”* (P10). The management of the concussion in the chronic phase generally meant that the injury had gone undiagnosed in the initial assessment of the injury: “*I got assessed for heart function, lung function, broken bones, but nothing for head, and it was only later that week I was dizzy. I was having headache, severe headaches, (I) couldn’t concentrate driving, driving at night”* (P11) and further supported by “*It was only driving back in the car afterwards, luckily there was a fella from my club coming back with me, because I just couldn’t, I couldn’t remember any of the game I had”* (P21). In some cases, after managing the chronic stages of concussion, the long-term symptomology was revealed once they had returned to light activity levels with their respective squads: *“All of a sudden, I was dizzy, nauseous. All the symptoms were back, and we were playing a practice match, and they (symptoms) were all back. I was just playing the match as normal, and then this came on me”* (P18) and *“they (management) are kind of able to tell me that like I’m OK, like not to panic, and because, I think I was just panicking about why do I feel this way”* (P16). 

## 4. Discussion

Gaelic games are a unique aspect of the sporting landscape in Ireland; they are an amateur organisation and have a tiered range of competitive levels within their games’ structures. This creates a natural two-tiered system in relation to amateur competitions (occurring within counties) and an elite amateur system, where the most talented athletes are selected to represent their region (county) in all-Ireland competitions on a league or championship basis. As Gaelic games are one of the most popular sports in Ireland, there are vast numbers of participants available to the GAA, regardless of the level of the sport being played [28]. Whether the squad is amateur or elite amateur, each athlete provides an enormous level of time and effort, on a voluntary basis, to these squads. 

With respect to the amateur tier of Gaelic games, many clubs are underfunded and are likely based in traditional rural areas of the country. In contrast, most elite amateur county (regional) organisations have sufficient resources available to them, at least for the senior male squads. It is apparent that many of the components required to be a competitive team are in place for these elite amateur athletes. They receive excellent coaching, have physiotherapy available to them, and other intangible benefits of being part of an inter-county squad. However, access to a suitable level of concussion education, management, and treatment does not seem to be a priority. This is a legitimate concern based on these qualitative interviews and the paucity of data available on true concussion incidence in Gaelic games. 

Based on the data acquired from this process, there are distinct measures that can be addressed to improve these existing shortcomings. Access to meaningful concussion education programmes should be considered for all stakeholders including players, coaches, referees, medical personnel, and allied health practitioners. These programmes need to be regular and certified to ensure that everyone is equipped to manage incidents of concussion effectively. Research has suggested that interventions are only effective once they are consistent, for example, delivering education programmes every playing season or twice a season [29,30]. There needs to be a set of standardised protocols across all levels of play; with respect to amateur levels of the game, there are concussion recognition tools available without the need of medically trained personnel [31,32]. With respect to elite amateur Gaelic games, there is an evident need to implement and enforce the use of the SCAT6 (or a similar tool) for concussion recognition and formalising a structured head injury assessment (HIA) protocol for the GAA. In tandem with these measures, a centralised and accurate recording of concussion incidence needs to be implemented to develop a true picture of the injury within Gaelic games settings [33]. It is standard practice for appropriate medical personnel to be in place for health and safety reasons at elite amateur levels. However, this is not the case at amateur levels whereby access to medical resources is limited or, in many cases, non-existent [5]. For these situations, clubs could implement basic first aid courses, train coaches to be vigilant for the signs and symptoms of concussion, and where there are any concerns, remove the athletes from the field of play [34]. This can be supported by follow-up conversations to check if the athlete has received further treatment. 

During the data analysis, it was obvious that a cultural shift is required around the issue of concussions in Gaelic games, supporting previous research by O’Connor et al. (2021) [11]. All the participants had experienced a concussion or multiple concussions firsthand, or they had observed teammates and competitors experience concussions. This implies that the issue of concussion in these sports may be widespread, and that clubs need to adopt a safety-first attitude [12]. These measures are relatively uncomplicated to initiate, but they require consistent efforts from the GAA to reinforce and ingrain these changes in attitudes. There needs to be a move away from the fear of stigma or repercussions about reporting concussions [35]. These will ultimately alleviate concerns that athletes may be experiencing and assist in the early identification and management of concussions. A final recommendation from this process is that further research needs to be initiated by the GAA into the prevalence of concussions in Gaelic sports, and the establishment of best practices for concussion management at all levels of Gaelic games. 

## 5. Conclusions and Limitations

The recognition and management of concussions in Gaelic games are currently inadequate and may be posing significant risks to athletes’ long-term health. The measures required to address these shortcomings requires a multifaceted approach involving education, standardised protocols, improved medical access, cultural change, and ongoing research. The significant limitation associated with this study is that the cohort size was limited to approximately twenty-five percent of the counties across the country. 

## Figures and Tables

**Table 1 healthcare-12-01974-t001:** Sample results of thematic analysis of interviews with current and retired Gaelic games athletes (n = 22).

Themes	Categories	Subcategories	Sample Quote
Inconsistent identification of concussion as an injury and the absence of standardised procedures for removal from play	Inconsistent adoption of head injury practices Club (amateur) structures and inter-county structures (elite amateur) Awareness and testing for concussion	Absence of standardised injury management Lack of action or positive action from medics and physios Non standardised baseline testing in elite amateurs	“You can see him (the player) on the edge of the camera, like the cameras following the ball, but you can see him on the edge, getting up, stumbling, falling, or they’re able to return to play, or running the wrong direction” (P5) “I was just down on the ground for a while my next memory, I suppose, is they didn’t take me off straight away, and I was still on the field, but I don’t really know how long, I still don’t know to this day how long I was kind of left there” (P9) “I say it’s very rare that a physio team or a management team would have taken off player because of that (concussion)” (P2) “Our medical team are extremely sharp; I’ve actually seen players being taken out against their own will and their own accord and so that’s been really good” (P4) “In general, it was no real awareness around it. I didn’t know that there was such a thing as long-term effects of it. Sure, you shake it off and you play on” (P9) “He (medical doctor) met me, do a few tests as in vision tests or the letters, a few memory tests......he might say a few letters, you just say them back....and if he was happy then with that, I was allowed to go back” (P22)
The impact of athlete-driven decision making on concussion management	Non-disclosure of injury incidence and symptoms Self-identification of concussion Athlete-led decisions on RTP	Influencing medics, non-disclosure, and being dismissive of symptoms Diagnosed and undiagnosed incidence of concussion Athletes determining length of RTP and self-managed recovery	“I wasn’t bad enough that they (management) thought that I needed to come off the field and so I managed to play the last ten minutes. Now I don’t know if I was fully there (cognitively present) for those ten minutes, I can’t really remember” (P21) “I was trying to explain this to someone yesterday because they texted me thinking they had a concussion is like you literally feel like you’re a fog over here, but you can’t explain that feeling” (P10) “I never got that (concussion) diagnosed or anything, but that’s what I had for a good while after, just a ringing in my ears that was stopping me sleeping. So like if I was to try put one broad term on it, just I just didn’t feel myself” (P18) “If I had decided I wanted to go back and play or train two- or three-days’ time, I don’t think anyone would have argued with me” (P13) “I had insomnia (due to symptoms) the week before, so I was getting very little sleep. I was like popping sleeping tablets like they were Skittles (sweets)” (P17)

## Data Availability

Data and materials related to the manuscript are available upon reasonable request from the corresponding author.

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
