# Peer review of "Athlete Perspectives on Concussion Recognition and Management in Gaelic Games: A Qualitative Analysis"

_healthcare, 2024, doi:10.3390/healthcare12191974_

Round 1

Reviewer 1 Report

Comments and Suggestions for Authors

This is a qualitative study using grounded theory methodology to explore perspectives on concussion recognition and management strategies of 22 Gaelic game athletes. The authors recruit athletes of both genders and different regions in Ireland. Overall, this is a well written and conducted study that is on track for acceptance. I only have minor comments below:

Minor comments

1.     Line 13: Please insert period after “making on concussion management”

2.     Line 46: Please remove the extra space in “2021)  .”

3.     Line 146: “The initial group of participants were identified via existing contacts…” Please elaborate on how subsequent participants were recruited. How did the authors recruit athletes from each respective region?

4.     Please clarify if the regions listed (E.g. Tyrone, Clare, etc) are in reference to the region that the athlete’s represent versus their place of origin. Also please clarify why this may be relevant to differentiate? Are there different regulatory protocols and procedures in each region which may influence athlete perspectives?

5.     Line 164-180: The “Participant characteristics” section can be moved to the results section

6.     Lines 281-285: Hard to interpret. The point made is that concussions are underdetected because of the provision of medical assistance? The quotation does not help clarify this.

7.     Lines 566-594: Great suggestions in general. The authors put the responsibilities on clubs to implement first aid, trained coaches, etc… However, this is already the case and its been demonstrated to be insufficient as an intervention leading to many inconsistent practices. Would the authors recommend the GAA or another governing body to take responsibility?

8.     Line 600: please remove the additional space in “need s”

9.     Overall the discussion is well written and thoughtful. There was effort by the authors to recruit male and females, but there is not much summarized on the differing perspectives if at all. Please add a few lines to clarify to readers.

Author Response

Sincere thanks for giving your time, advice and suggestions to review and improve this article.

1.Line 13: Please insert period after “making on concussion management”

Response – thanks for the observation, period has been inserted

2.Line 46: Please remove the extra space in “2021).

Response – the extra space has been removed”

3.Line 146: “The initial group of participants were identified via existing contacts…” Please elaborate on how subsequent participants were recruited. How did the authors recruit athletes from each respective region?

Response – We used a snowball sampling methodology for this research paper whereby an initial sample of participants were identified via existing research team contacts. When participants were recruited / interviewed, they were asked to suggest other athletes whom they felt may be willing to take part in the research (participants were asked to suggest additional participants but they were not obliged to do so and supplied names/contact details of their own free will). The regional aspect was random, we did not try to select from specified areas.

4.Please clarify if the regions listed (E.g. Tyrone, Clare, etc) are in reference to the region that the athlete’s represent versus their place of origin. Also please clarify why this may be relevant to differentiate? Are there different regulatory protocols and procedures in each region which may influence athlete perspectives?

Response – thanks for the comment, the nature of Gaelic games is that you can only represent the area you originate, it’s rare that a person from a specific region (e.g., Clare) would represent another region (e.g., Cork). All the participants are non-paid players and as such, they have an obligation to represent their ‘home’ region. There are no differences in regulatory processes in each region which may influence the athletes’ perspectives; the regions were listed as this represents a sample of approximately 25% of the total number of regions (counties) on the island of Ireland.

5.Line 164-180: The “Participant characteristics” section can be moved to the results section

Response – thanks for the suggestion, we as a research team are opting to keep this section in the ‘methods’ section for the purposes of continuity in the transcript.

6.Lines 281-285: Hard to interpret. The point made is that concussions are under detected because of the provision of medical assistance? The quotation does not help clarify this.

Response – sincere thanks for this observation, an edit has been made to clarify the point and connect it to the quotation.

7.Lines 566-594: Great suggestions in general. The authors put the responsibilities on clubs to implement first aid, trained coaches, etc… However, this is already the case and its been demonstrated to be insufficient as an intervention leading to many inconsistent practices. Would the authors recommend the GAA or another governing body to take responsibility?

Response – thanks for the comment, the research team have begun the process of lobbying the GAA and the Irish government (Minister for Sport) based on this current research. We mention this as a suggestion in lines 610-612

8.Line 600: please remove the additional space in “need s”

Response – thanks for the observation, the additional space has been removed

9.Overall the discussion is well written and thoughtful. There was effort by the authors to recruit male and females, but there is not much summarized on the differing perspectives if at all. Please add a few lines to clarify to readers.

Response – thanks for the suggestion; overall the experiences were very similar for male and female athletes in terms of concussion experiences. The major differences (which may be address in a subsequent paper) were around access to medical care and funding for elite amateur male and elite amateur females athletes.

Reviewer 2 Report

Comments and Suggestions for Authors

Overall, the topic of concussion recognition and management focuses on an interesting national sporting system and furthers this important discussion related to protecting sports participants at all levels.  The comparison between elite and amateur sports, as well as differences in treatment by gender, is valuable.  The sample of former athlete participants is strong and representative.  The qualitative findings seem authentic and representative of the participant data and contribute to a good discussion that addresses the authors' main research questions.  Including a table to represent themes and subthemes along with the participant quotes provides good emphasis on the main findings.  

Below are some minor recommended edits: 

Methods:

I suggest providing more detail about the central questions asked or addressed in the interviews.   I suggest either listing the interview questions in this section or giving more specific details about the interview guide as this will help the reader to better understand the results and conclusions.  

Regarding the interview participants who are retired, I suggest a brief statement about how long they have been retired.  If your purpose is to provide a "current" understanding of concussion, then including this information will help the reader understand the currency of your participant responses. 

Discussion:

I suggest some small edits in the formatting of your discussion narrative to be consistent with Table 1.  Table 1 provides clear themes, categories, and subcategories of the qualitative data.  Your narrative could be more clear in identifying these same levels.  One simple example would be to include a summary statement that lists categories and subcategories along with each of your themes in sections 3.1 and 3.2.  This would be helpful for the reader since it is a bit confusing to discern between themes and categories in the narrative. 

Regarding participant quotes, it would be effective to indent and separate longer quotes (more than 40 words) if the manuscript guidelines allow this.  Also, most (all?) of the quotes are included within the paragraph which can deemphasize their importance and seem to run together.  The interview data is interesting and relevant for the reader, so consider how you can isolate key quotes to emphasize them and strengthen the flow and impact of the discussion.

Author Response

Sincere thanks for giving your time, advice and suggestions to review and improve this article.

Methods:

I suggest providing more detail about the central questions asked or addressed in the interviews.   I suggest either listing the interview questions in this section or giving more specific details about the interview guide as this will help the reader to better understand the results and conclusions. 

Response – sincere thanks for the comment, we will address this is the ‘data availability’ statement i.e., accommodating reasonable requests for data and supplemental materials such as the interview guide.

Regarding the interview participants who are retired, I suggest a brief statement about how long they have been retired.  If your purpose is to provide a "current" understanding of concussion, then including this information will help the reader understand the currency of your participant responses.

Response – thanks for the comment, 18 participants remain actively involved and we believe this is reflective of the current understanding. On lines 179-180, we have included data about those participants who’ve had to retire due to concussion. There was a large degree of variance in terms of what participants believed to be retired i.e., those who had to retire from their Gaelic games code due to concussion were still active in playing non-contact sports.

Discussion:

I suggest some small edits in the formatting of your discussion narrative to be consistent with Table 1.  Table 1 provides clear themes, categories, and subcategories of the qualitative data.  Your narrative could be more clear in identifying these same levels.  One simple example would be to include a summary statement that lists categories and subcategories along with each of your themes in sections 3.1 and 3.2.  This would be helpful for the reader since it is a bit confusing to discern between themes and categories in the narrative.

Response – sincere thanks for the suggestion, a comment has been inserted at the beginning of the results section to direct readers to Table 1 for clarification on categories and sub categories.

Regarding participant quotes, it would be effective to indent and separate longer quotes (more than 40 words) if the manuscript guidelines allow this.  Also, most (all?) of the quotes are included within the paragraph which can deemphasize their importance and seem to run together.  The interview data is interesting and relevant for the reader, so consider how you can isolate key quotes to emphasize them and strengthen the flow and impact of the discussion.

Response – sincere thanks for this helpful observation, the authors have reviewed the script to indent all quotes over 40 words in length.

Reviewer 3 Report

Comments and Suggestions for Authors

The name of the organization that provided ethical approval for the research is missing. Furthermore, I kindly ask the authors to address why a larger sample of participants was not selected.

Author Response

Sincere thanks for giving your time, advice and suggestions to review and improve this article.

The name of the organization that provided ethical approval for the research is missing.

Response – Thanks for the comment, the university name has now been inserted

 Furthermore, I kindly ask the authors to address why a larger sample of participants was not selected.

Response – a larger sample size was not recruited as data saturation had occurred when we reached a sample size of 22 participants. We believed this was a suitable sample size in relation to qualitative research